# *Brassicaceae* transcriptomes reveal convergent evolution of super-accumulation of sinigrin

Jinghua Yang [1,2], Zhangping Li[1], Jinmin Lian[3], Guoning Qi[4], Pibiao Shi[5], Jiawei He[6], Zhongyuan Hu[1] & Mingfang Zhang[1,2] ✉

Wasabi, horseradish and mustard are popular pungent crops in which the characteristic bioactive hydrolysis of specialized glucosinolates (GSLs) occurs. Although the metabolic pathways of GSLs are well elucidated, how plants have evolved convergent mechanisms to accumulate identical GSL components remains largely unknown. In this study, we discovered that sinigrin is predominantly synthesized in wasabi, horseradish and mustard in *Brassicaceae*. We de novo assembled the transcriptomes of the three species, revealing the expression patterns of gene clusters associated with chain elongation, side chain modification and transport. Our analysis further revealed that several gene clusters were convergently selected during evolution, exhibiting convergent shifts in amino acid preferences in mustard, wasabi and horseradish. Collectively, our findings provide insights into how unrelated crop species evolve the capacity for sinigrin super-accumulation and thus promise a potent strategy for engineering metabolic pathways at multiple checkpoints to fortify bioactive compounds for condiment or pharmaceutical purposes.

[1] Laboratory of Germplasm Innovation and Molecular Breeding, Institute of Vegetable Science, Zhejiang University, 310058 Hangzhou, China. [2] Key Laboratory of Horticultural Plant Growth and Development, Ministry of Agriculture and Rural Affairs, 310058 Hangzhou, China. [3] Biozeron Shenzhen, Inc., 518081 Shenzhen, China. [4] State Key Laboratory of Subtropical Silviculture, Zhejiang A&F University, Lin'an, 311300 Hangzhou, China. [5] Xinyang Agricultural Experiment Station of Yancheng, 224049 Yancheng, China. [6] Alpine Economic Plant Research Institute of Yunnan Academy of Agricultural Sciences, 674199 lijiang, China. ✉email: mfzhang@zju.edu.cn

Plants synthesize a multitude of compounds that help them adapt to their ecological niches, but also increase their popularity as a source of food for humans[1]. Glucosinolates (GSLs) are specialized metabolites in the family *Brassicaceae*[2,3] that act as protectants against generalists[4,5], and can mediate the coevolutionary association between herbivores and plants. The bioactive products of GSL hydrolysis, typically isothiocyanates and nitriles[6,7], are associated with improved health benefits[7]. Well-known allyl-isothiocyanate or benzyl-isothiocyanate (ITC) is a characteristic determinant of the pungent flavor associated with wasabi. The products of GSL hydrolysis, especially sinigrin-derived ITCs, are major determinants of the pungent aroma of wasabi (*Wasabi japonica*) roots, horseradish (*Armoracia rusticana*) roots and mustard (*Brassica juncea*) seeds[8–10], which release ITC when grated[11]. Moreover, the breakdown of sinigrin contributes to the characteristic taste of mustard[12,13]. Progress has been made over the last two decades with regard to creating a model ecological system for identifying causal genes underlying natural variation, and this model features the biosynthesis of GSLs and both storage and transport pathways[3]. Understanding the evolutionary and genetic bases of such metabolic diversity and specificity, could provide crucial clues for chemical ecology and yield useful tools for crop breeding. Such an understanding could also create opportunities for producing and engineering plant-derived chemicals for the biofortification of certain pharmaceutical applications.

Convergent evolution is the appearance of similar phenotypes in distinct evolutionary lineages[14,15], and is frequently observed for specialized metabolites in plants[1,16,17]. A particularly intriguing goal in evolutionary biology is to understand the genetic mechanisms, that result in different species exhibiting the same levels of variation in specialized metabolites. A large number of genetic and biochemical mechanisms could give rise to convergent evolution. Moreover, the convergent evolution of specialized metabolism in plants is surprisingly common and predominantly involves the same product from the same substrate, but with unrelated enzymes and genes or the same product from different substrates[1].

The synthesis of GSLs is emerging as a unique model with which to investigate nonlinear selection and a blend of convergent, divergent, and parallel evolution that shapes natural variation[7,18]. However, we do not yet fully understand how the selectivity and evolution of GSL transport may underlie natural variation within and between species. Early studies showed that de novo biosynthesis does not occur in seeds[19], and it is generally assumed that GSLs are synthesized along the vasculature in leaves and subsequently transported from the leaves to the seeds[20]. Recent studies demonstrated that two *Arabidopsis* transporters, AtNPF2.10 (AtGTR1) and AtNPF2.11 (AtGTR2), are responsible for transporting GSLs, particularly long-chain aliphatic GSLs[21–23]. In roots, GTR-mediated import is essential for the retention of GSLs in the *gtr1gtr2* mutant, and GSLs are moved from root to shoot[24]. In contrast, short-chain aliphatic GSLs have been shown to be mainly produced in the leaves and then transported to the roots[24]. Furthermore, GTR1, GTR2, and GTR3 (AtNPF2.9) all contribute to distributing indole GSLs between roots and shoots[25].

In general, a large amount of diversity exists in both the amount and profile of GSLs in *Brassicaceae* crops[26], thus resulting in a range of different flavors. The diverse components that make up GSLs are thought to need different transporters in *Brassicaceae* crops and the model plant *Arabidopsis*. However, we have yet to determine how plant lineages from families that are distantly related to *Brassicaceae* have evolved mechanisms to accumulate GSLs. In the present study, we used comparative transcriptome analysis to dissect the major causal loci and gene clusters underlying the convergent evolution of GSL accumulation.

## Results

**Comparative analyses of GSLs in wasabi, horseradish, and mustard.** We constructed a phylogenetic tree for wasabi, horseradish and mustard species with several other sequenced species using one-to-one orthologs in the *Brassicaceae* family. The wasabi, horseradish, and mustard species all belonged to different lineages within the family (Fig. 1a). According to the definition of convergent evolution, the accumulation of sinigrin in these three species is considered a result of convergent evolution, because similar phenotypes are evident in distinct evolutionary lineages within the same family[15]. We determined GSL components in the leaves, roots, and seeds using high-performance liquid chromatography (HPLC). The content of aliphatic GSLs was much more intensively enriched than the content of indolic GSLs in the leaves, roots, and seeds of these three species (Supplementary Fig. 1 and Supplementary Data 1). 3C-aliphatic-sinigrin is a predominant component in the leaves of all of the species (Fig. 1b and Supplementary Data 2). 3C-aliphatic-sinigrin accumulates far more extensively in the harvestable roots of wasabi and horseradish than in mustard (Fig. 1b and Supplementary Data 2). We observed that wasabi and mustard seeds mainly accumulate sinigrin, while horseradish seeds accumulate a greater proportion of gluconapin (Fig. 1b and Supplementary Data 2). GSLs have been shown to be synthesized in various tissues, including the root, leaf, stem, and silique wall, and then translocated from source to sink[20,21,25]. These results suggested the convergent accumulation of sinigrin in these three species.

**The de novo assembly of transcriptomes for wasabi, horseradish and mustard.** We de novo assembled the transcriptomes of *W. japonica*, *A. rusticana*, and *B. juncea*. Twenty-seven samples of leaves, roots, and seeds were collected in triplicate for Illumina sequencing. Following data assessment, quality control, filtration, and assembly, we obtained an average of 5.46 Gb of clean data for each sample. This generated 53,660, 61,283, and 69,279 unigenes for these three species, respectively (Table 1, Supplementary Data 3–5, and Supplementary Fig. 2). We estimated the transcriptome assemblies using BUSCO, in which over 50% of the gene models were annotated for the three species (Supplementary Data 6). In total, 64,774 common transcripts were shared across the three transcriptomes, and specific transcripts were identified in all three species (Fig. 2a). The biological replicates used for transcriptome sequencing of the leaves, roots, and seeds of these three species were validated using principal component analysis. These analyses indicated that the expression patterns showed high levels of similarity between biological replicates (Fig. 2b).

We identified genes related to the biosynthesis, transport, and breakdown of GSLs based on the metabolic pathways of GSLs from the three species and their orthologs from *Arabidopsis* (Supplementary Data 7). Next, we investigated the transcriptional expression of genes involved in the metabolic pathways of GSLs, including side chain modifications, transcription factors, and transporters within the NPF family (Fig. 2c–f and Supplementary Fig. 3 and Supplementary Data 8). Analysis indicated that some identical functional genes (*IGMT1*, *FMOGS-OX1*, *FMOGS-OX4*, and *CYP81F1*) showed similar transcriptional patterns in the leaves of all species but also revealed that several diverse genes (*GSL-OH*, *FMOGS-OX3*, and *CYP81F4*) were expressed at similar levels in the leaves of

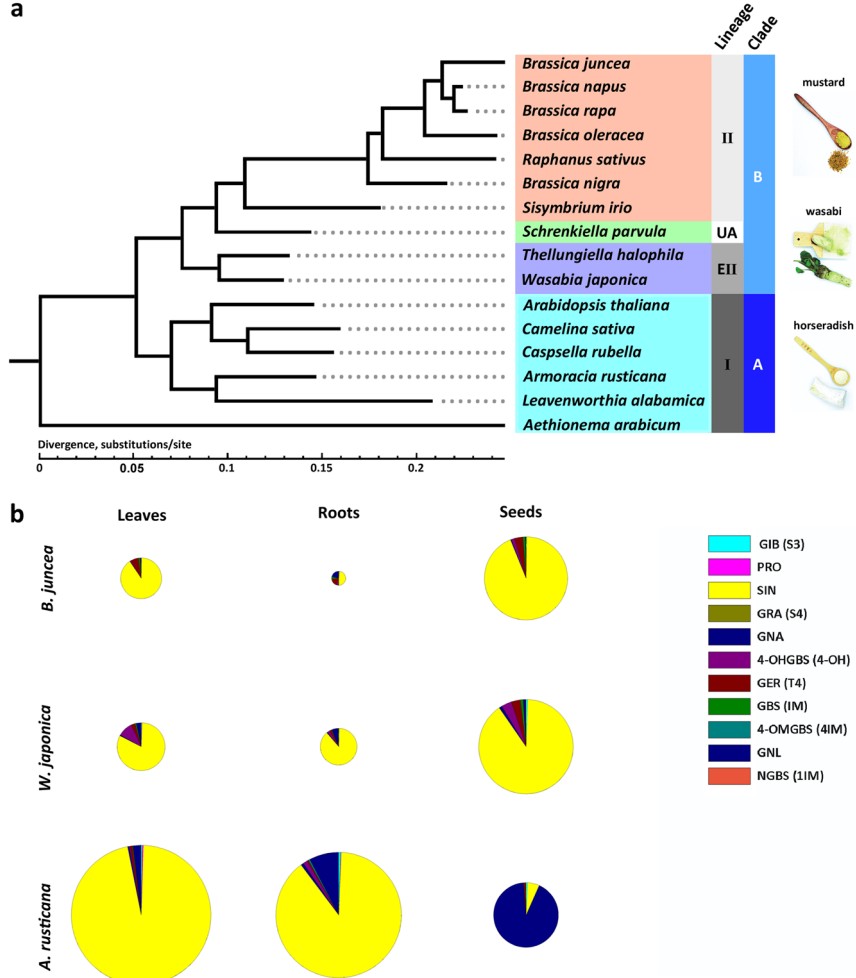

**Fig. 1 GSL components in the leaves, roots and seeds of *W. japonica*, *A. rusticana* and *B. juncea*. a** Phylogenetic tree of *Brassicaceae* and three species with *wasabi* qualities (mustard, *B. juncea*; wasabi, *W. japonica*; and horseradish, *A. rusticana*). I, II and EII indicate Lineage I, II, and Expanded Lineage II, respectively. A and B refer to clades. **b** GSL components in the leaves, roots, and seeds of *B. juncea*, *W. japonica*, and *A. rusticana*.

the two species (Fig. 2d–f). However, we found that some genes involved in the modification of GSL side chains were expressed at higher levels in roots, suggesting that GSLs might also be synthesized in the roots (Fig. 2d). Previous research also demonstrated that GSLs, particularly indole GSL, can be synthesized in both the leaves and the roots[27]. We confirmed that *AOP2* was expressed at significantly higher levels in leaves than in seeds and roots (Supplementary Fig. 4), in which this gene plays a key role in the biosynthesis of glucosinolates that then convert glucoiberin to sinigrin. With regard to the transcriptional regulation of GSL biosynthesis, we found that the transcription factor *Dof1.1* exhibited the highest expression levels in all three species and that the transcription factors *MYB28* and *IQD* exhibited higher expression levels in two of the species (Fig. 2e). The higher expression levels of genes in the NPF family in leaves and roots of the three species suggested that these genes are potential transporters of GSLs from sink to source (Fig. 2f).

**Phylogenetic evolutionary analysis of wasabi, horseradish, and mustard.** We then collected single-copy genes from the A subgenomes of *B. juncea* and *B. napus* and from *A. thaliana*, *A. rusticana*, *B. oleracea*, *B. rapa*, and *W. japonica* in *Brassicaceae*. Using these single-copy-gene orthologs, we constructed a

phylogenetic tree showing independent evolutionary lineages for *B. juncea*, *A. rusticana*, and *W. japonica* (Fig. 3a). Sinigrin is specifically enriched in several species of *Brassicaceae*, including *B. nigra*, *B. juncea*, and *B. carinata*, but not in *B. rapa*[26] and several species in other *Brassicaceae* clusters, including *W. japonica* and *A. rusticana*[10]. We calculated the Ka and Ka/Ks (nonsynonymous/synonymous) values of all single-copy genes and GSL-related genes in these seven species. The Ka values of *B. juncea*, *A. rusticana*, *W. japonica*, and *A. thaliana* were much higher than those of *B. rapa*, *B. oleracea*, and *B. napus* (Fig. 3b and Supplementary Fig. 5), suggesting parallel selection within these species. We also used positively selected gene (PSG) analysis to identify candidate selected genes in *A. rusticana*, *B. juncea* and *W. japonica*. *A. rusticana*, *B. juncea*, and *W. japonica* were used as the foreground branches while *B. napus*, *B. oleracea*, *B. rapa*, and *A. thaliana* were used as background branches in the PSG analysis. This analysis showed that 354 genes were under positive selection in *B. juncea*, *A. rusticana*, and *W. japonica* (Supplementary Data 9).

**Convergent evolution analysis in wasabi, horseradish, and mustard.** Next, we used the PCOC package, which allows for the accurate detection of convergent shifts in amino acid preferences[28], to identify nine genes exhibiting convergent

**Table 1 De novo assembly of transcriptomes of *W. japonica*, *A. rusticana* and *B. juncea*.**

| | W. japonica | | | | A. rusticana | | | | B. juncea | | | |
|---|---|---|---|---|---|---|---|---|---|---|---|---|
| | Unigene | | Transcripts | | Unigene | | Transcripts | | Unigene | | Transcripts | |
| | Total number | Percentage (%) | Total number | Percentage (%) | Total number | Percentage (%) | Total number | Percentage (%) | Total number | Percentage (%) | Total number | Percentage (%) |
| 300-500 | 19,024 | 35.45 | 44,418 | 24.27 | 21,907 | 35.75 | 65,804 | 28.97 | 21,324 | 30.78 | 65,998 | 20.99 |
| 500-1000 | 16,333 | 30.44 | 61,058 | 33.36 | 21,920 | 35.77 | 93,992 | 41.38 | 23,108 | 33.35 | 115,261 | 36.66 |
| 1000-2000 | 12,511 | 23.32 | 54,235 | 29.63 | 14,190 | 23.15 | 57,177 | 25.17 | 18,358 | 26.50 | 101,617 | 32.32 |
| 2000+ | 5792 | 10.79 | 23,341 | 12.75 | 3266 | 5.33 | 10,197 | 4.49 | 6,489 | 9.37 | 31,493 | 10.02 |
| Total number | 53,660 | | 183,052 | | 61,283 | | 227,170 | | 69,279 | | 314,369 | |
| Total length | 53,183,810 | | 204,136,944 | | 52,378,102 | | 197,708,225 | | 68,610,536 | | 338,263,682 | |
| N50 length | 1407 | | 1485 | | 1081 | | 1049 | | 1322 | | 1350 | |
| Mean length | 991 | | 1115 | | 855 | | 870 | | 990 | | 1076 | |

evolution in *B. juncea*, *A. rusticana*, and *W. japonica*, using *B. rapa* as a control. These genes are involved in the biosynthesis (*CYP79A1*, *CYP83A1*, *GSTF11*, *SUR1*, and *UTG74B1*), transcriptional regulation (*Dof1.1* and *IQD*) and breakdown (*NIT1*) of GSLs (Fig. 4 and Supplementary Data 10). The amino acids encoded by these genes displayed convergent shifts in *B. juncea*, *A. rusticana*, *W. japonica*, and *A. thaliana* but showed divergent variations in *B. rapa*, *B. oleracea*, and *B. napus* (Fig. 4). For example, the amino acids (Q233L and K364N) in the SUR1 gene are involved in the core structural formation of GSLs from *S*-alkyl-thiohydroximate to thiohydroximate and showed convergent substitutions in species known to accumulate large amounts of sinigrin (*B. juncea*, *A. rusticana*, and *W. japonica*) compared to other species (*B. rapa*, *B. oleracea*, and *B. napus*) in the phylogenetic tree (Fig. 4). In addition, several other amino acids (74th, 122nd, and 185th) showed convergent shifts in *B. juncea*, *A. rusticana*, and *W. japonica* compared to *B. rapa*, *B. oleracea*, and *B. napus* in the phylogenetic tree (Fig. 4). These results suggested that the convergent evolution of genes associated with the metabolism of GSLs may contribute to the convergent enrichment of sinigrin in *B. juncea*, *A. rusticana*, and *W. japonica*.

## Discussion

The process of convergent evolution, in which distinct lineages independently evolve similar traits, has fascinated evolutionary biologists for centuries[29], predominantly because convergent evolution is generally thought to represent a visible manifestation of the power of selection. The plant and animal kingdoms evolve similar traits under selection in order to adapt to different ecological environments. However, most of the existing research on convergent or parallel evolution is focused on morphological variations, and very little is known about metabolic pathways and the release of bioactive metabolites in *Brassicaceae*. Previous studies on the evolution of *Brassicaceae* indicated the prevalence of parallel or convergent evolution of several traits over time across the entire family[30]. Indeed, recent publications have reported that several specific traits, including tetraterpene biosynthesis[31], leaf head formation[32], and flowering time[33], have undergone parallel or convergent domestication or selection.

With the development of high-throughput sequencing technologies, numerous research studies have begun to investigate the specific genetic mechanisms, that underlie convergent traits and to test whether and under which conditions phenotypic convergence is associated with convergence at the genetic level[33]. Prior to the availability of genomic sequences, the transcriptomes of orphan species were of benefit in the identification of genes involved in GSL metabolism. Plants in the *Brassicaceae* family are now becoming an attractive model for studying the evolution of bioactive compounds. This is because these plants contain an abundance of GSLs and because their biosynthetic pathways are well documented. In the present study, we used de novo transcriptome assembly and PSG and PCOC analyses to identify the genetic mechanisms associated with the convergent evolution of sinigrin super-accumulation in mustard, wasabi, and horseradish. We found that the evolution of aliphatic GSLs is associated with multiple genes in a diverse array of pathways, and that the natural variations in alkenyl GSLs are caused by parallel nonfunctional alleles in the conserved *AOP2* locus. As exemplified by the *GS-OH* locus, the three different lineages of *Brassicaceae* most likely evolved the same components of GSLs by coopting different enzymes. Furthermore, it was found that the *Elong* locus has been subject to a complex

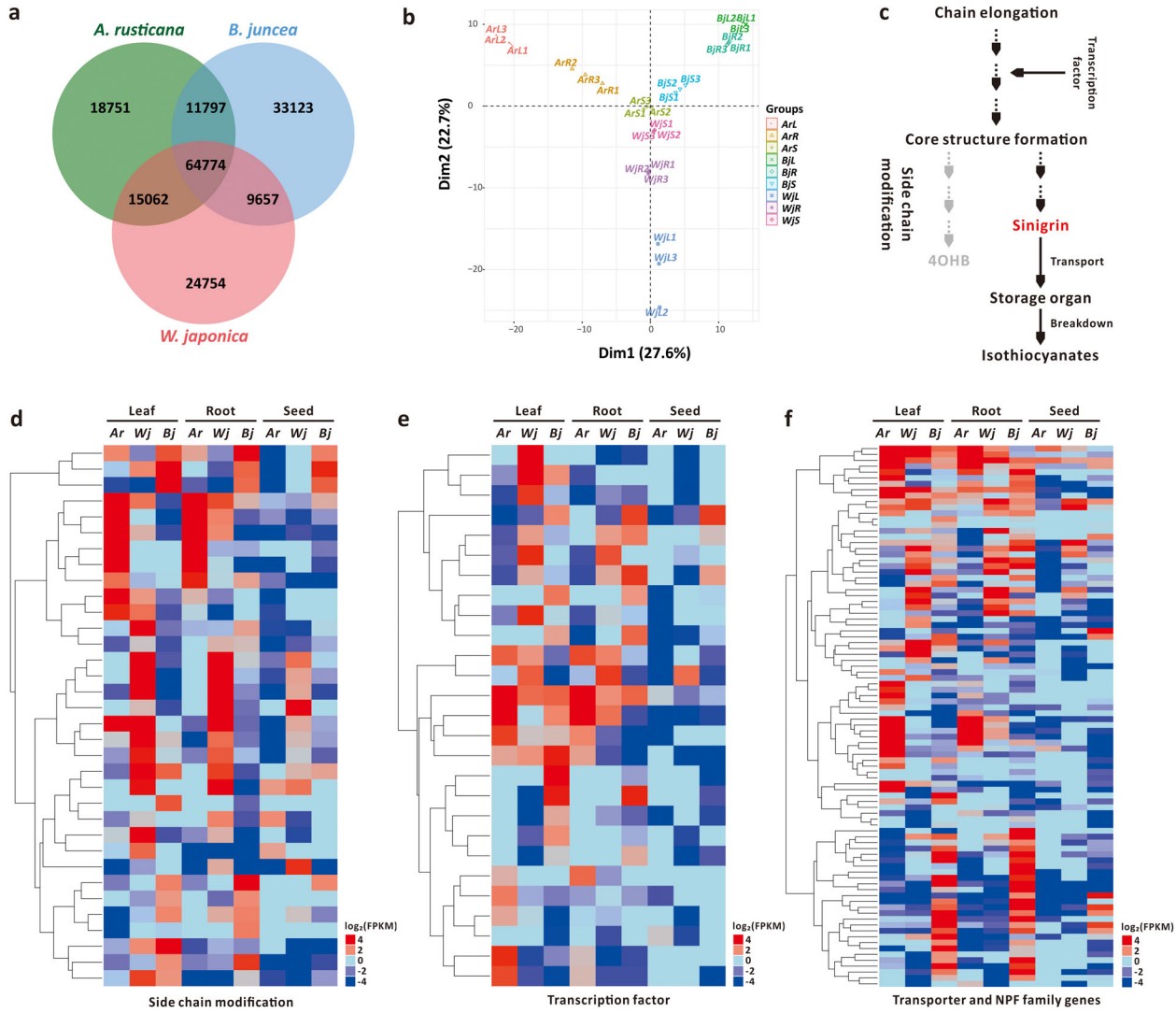

**Fig. 2 De novo transcriptome assemblies and transcriptional expression levels of genes related to GSL metabolism in *W. japonica*, *A. rusticana*, and *B. juncea*. a** Venn diagram of the transcriptomes of *W. japonica*, *A. rusticana*, and *B. juncea*. **b** Principal component analysis of the transcriptomes from the leaves, roots and seeds of *W. japonica*, *A. rusticana*, and *B. juncea*. **c** Schematic pathway of the biosynthesis, transport and breakdown of sinigrin. **d** Expression levels of genes related to the side chain modifications of GSL biosynthesis. **e** Expression levels of genes involved in the transcriptional regulation of GSL biosynthesis. **f** Expression levels of genes related to the transport of GSLs and in the NPF family.

combination of divergent and convergent evolution[18]. In this study, we found that convergent shifts of amino acids were largely imposed on genes involved in the formation of core structures and transcriptional regulation of GSLs. These observations imply that selection on these genes in core pathways leads to convergent evolution for the super-accumulation of sinigrin-ITCs in three distantly related species of *Brassicaceae*. This finding provides insights into how to select convergent GSL biosynthetic targets via metabolic engineering in order to enhance the synthetic capacity of leaf sources and improve certain traits in *Brassicaceae*. This is important because sinigrin is present in a range of vegetables within the *Brassicaceae* family, including broccoli, Brussels sprouts, black mustard seeds, and wasabi.

Metabolic gene clusters that are associated with specialized metabolism in plants represent the genomic signatures for metabolic evolution, define the specific features of plants and provide us with options for exploiting such plants with regard to their synthetic biological applications[34]. By investigating

gene clusters associated with GSL diversity in *Brassicaceae*[35], it is possible to enhance knowledge of taxonomy and identify plants with greater agricultural potential and abiotic stress tolerance[36]. Further studies are now needed to investigate whether these transporter genes are clustered in genomic regions with coregulatory mechanisms. In the present study, we used the comparative transcriptomes of three species of *Brassicaceae* to elucidate the evolutionary trajectories of crucial bioactive compounds formed from GSLs. Common selection on genes involved in biosynthetic and transport pathways plays a key role in the super-accumulation of sinigrin. Our findings will further advance our understanding of convergent evolution in relation to how plants super-accumulate specific bioactive compounds for improving environmental fitness. Moreover, our work opens up the possibility of developing toolkits to engineer metabolic pathways, so that we can use synthetic biology to improve the synthesis of bioactive compounds for condimental or pharmaceutical purposes.

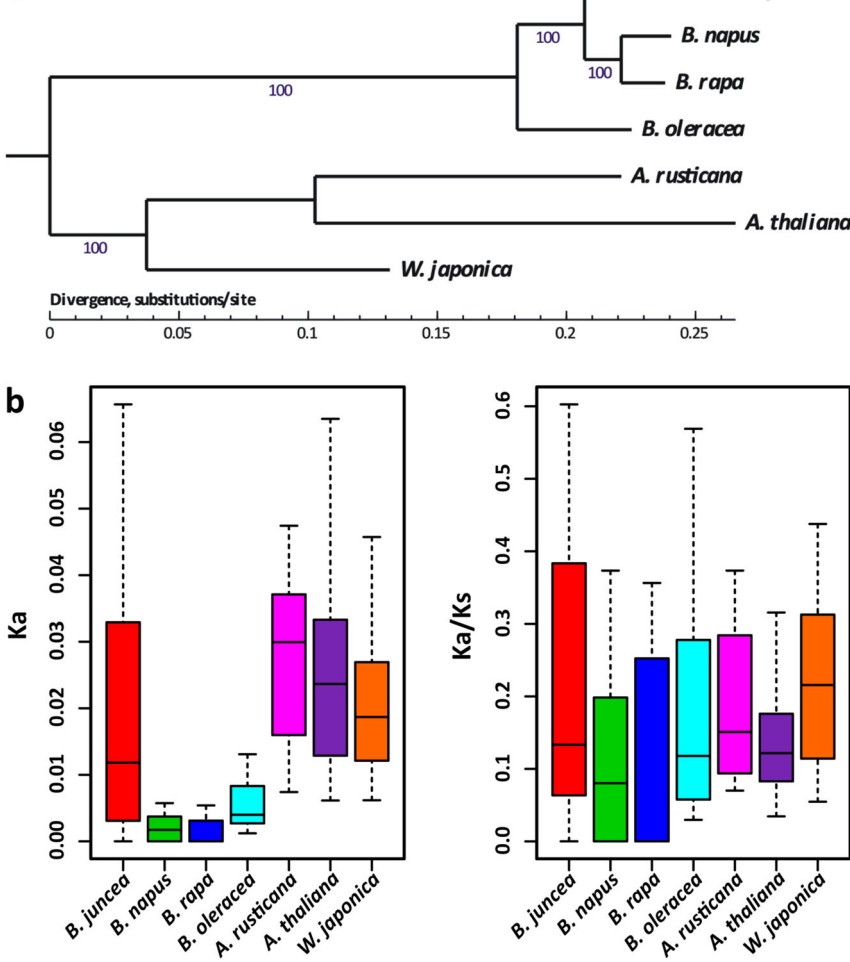

**Fig. 3 Phylogenetic evolutionary analysis of genes related to GSL metabolism in *Brassicaceae*. a** Phylogenetic tree of *W. japonica*, *A. rusticana*, and *B. juncea* along with several other *Brassicaceae* species. **b** Ks and Ka/Ks values for seven species of *Brassicaceae* obtained using one-to-one GSL metabolism-related orthologs in *Brassicaceae*.

## Methods

**Plant materials and GSL component assays**. Three *Brassicaceae* species were used to study the convergent evolution of wasabi quality: wasabi (*Wasabi japonica*), horseradish (*Armoracia rusticana*), and mustard (*Brassica juncea*). We collected leaves, roots and seeds from all three species for transcriptome analysis. The leaf and root samples were collected when the root organs had formed, and the seeds were collected when mature in the population. GSL profiles were analyzed by HPLC in accordance with standard protocols[37].

**De novo transcriptome assemblies of *B. juncea*, *A. rusticana*, and *W. japonica*.** Total RNA was extracted from the leaves, roots, and seeds of the three species using TRIzol Reagent (Life Technologies, California, USA). We then constructed a cDNA library for each sample according to the instruction manual (Illumina, USA) and sequenced PE150 fragments using an Illumina HiSeq™ 2500 sequencing platform (Illumina, USA) with a standard pipeline. Quality control measures were then applied, and the Trinity pipeline was used to create de novo assemblies from the clean data[38] with the following parameter settings: min_glue = 3, $V$ = 10, edge-thr = 0.05, min_kmer_cov = 3, path_reinforcement_distance = 85, group_pairs_distance = 250. All other parameters were set as the default. Next, we removed any redundant fragments using TGICL (TGI Clustering tools) and the Phrap assembler[39]. The following parameters were then used to ensure that the assemblies were of high quality: a minimum identity of 95%, a minimum of 35 overlapping bases, a minimum of 35 scores, and a maximum of 25 unmatched overhanging bases at sequence ends.

**Calculation of Ka and Ks**. Based on a phylogenetic tree of seven plant species (*A. thaliana*, *A. rusticana*, *B. juncea*, *B. napus*, *B. oleracea*, *B. rapa*, and *W. japonica*) and one-to-one orthologous genes, we estimated the evolutionary rates (Ka and Ks) of each branch using the CodeML tool in the PAML package (version 4.9 h) and

the free-ratio "branch" model[40], which allows distinct evolutionary rates for the branches.

**Positive selection analysis**. Based on the phylogenetic tree created for the three species of *Brassicaceae*, we used the branch-site model incorporated in the PAML package to detect positively selected genes (PSGs). The null model used in the branch-site test assumed that the Ka/Ks values for all codons on all branches were ≤1, whereas the alternative model assumed that the foreground branch included codons evolving with a Ka/Ks >1. A maximum likelihood ratio test (LRT) was used to compare the two models. The $p$ value was calculated according to the chi-square distribution with 1 degree of freedom (df = 1). Then, the $p$ values were adjusted for multiple testing using the false discovery rate (FDR) method. Genes were identified as being positively selected when the FDR < 0.05. Furthermore, we required that at least one amino acid site possess a high probability of being positively selected (Bayes probability >95%). If none of the amino acids passed this cutoff in the PSG analysis, then these genes were identified as false positives and excluded.

**Convergent evolution analysis**. One-to-one orthologs were identified in seven *Brassicaceae* species (*A. thaliana*, *A. rusticana*, *B. juncea*, *B. napus*, *B. oleracea*, *B. rapa*, and *W. japonica*). To detect the convergent evolution of genes among *A. rusticana*, *B. juncea*, and *W. japonica*, we used the Profile Change with One Change (PCOC) package with a posterior probability threshold of 0.95 to detect convergent amino acids for each gene[28]. Because *B. rapa* does not accumulate sinigrin[26], we further required at least one candidate convergent amino acid for *A. rusticana*, *B. juncea*, and *W. japonica* that was different from that in *B. rapa*.

**Statistics and reproducibility**. All RNA-sequencing data were from three independent biological replications. The GSL components and gene expression analysis were performed with three independent biological replications in this study.

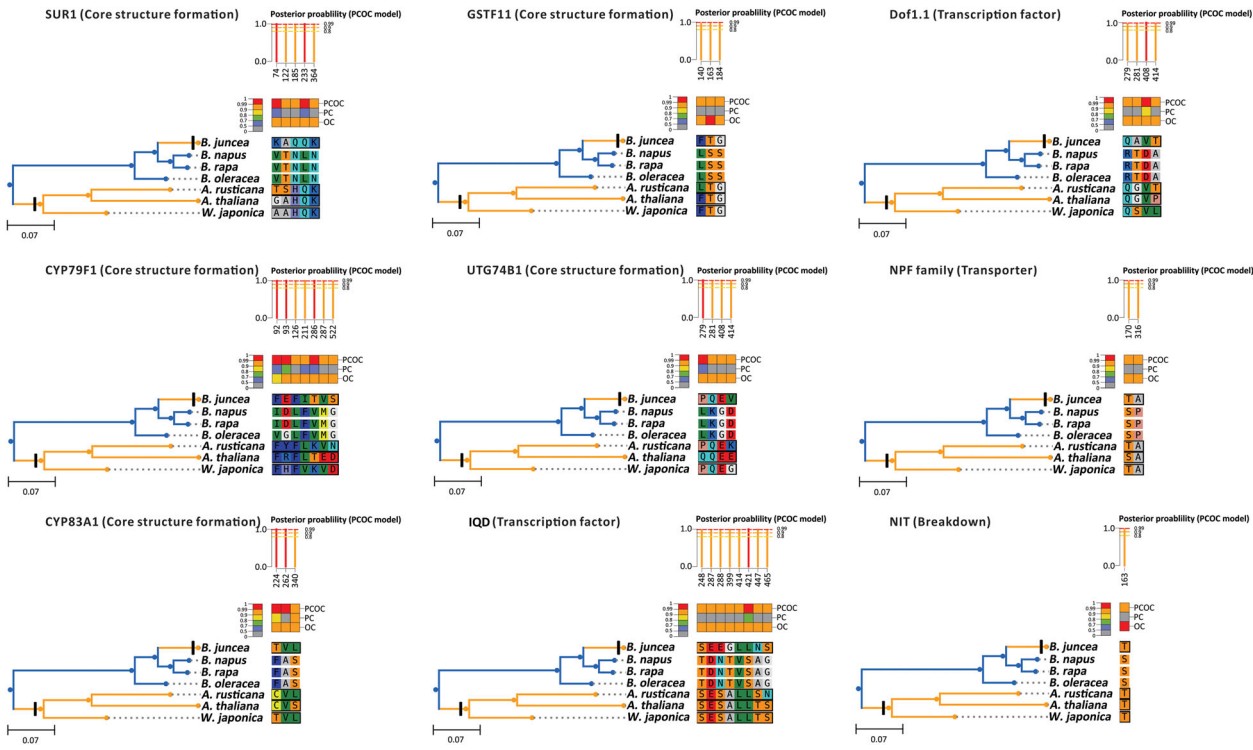

**Fig. 4 Convergent evolution analysis of genes related to GSL metabolism in *Brassicaceae*.** Detection of convergent shifts in the SUR1, Dof1.1, and NPF protein families in *Brassicaceae* using the PCOC toolkit. Sites are ordered by their posterior probability (pp) of being convergent according to the PCOC model, and sites are numbered according to amino acid sequence. Posterior probabilities for the PCOC, PC, and OC models are represented by different colors: pp ≥ 0.99 (red), pp ≥ 0.9 (orange), pp ≥ 0.8 (yellow), and pp < 0.8 (gray).

**Reporting summary**. Further information on research design is available in the Nature Research Reporting Summary linked to this article.

## Data availability

All the transcriptome data, including those for unigenes and differentially expressed genes, generated in the present study have been deposited in NCBI (https://www.ncbi.nlm.nih.gov/geo/) under accession number PRJNA670607, and in CNSA (https://db.cngb.org/cnsa/) under accession number CNP0001119.

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

## Acknowledgements

The authors thank Dr. H. Miao for GSL component analyses. This work was supported by the National Natural Science Foundation of China (32030092, 31872095), the National Natural Science Foundation of Zhejiang Province (LZ20C150002), and the Cheung Kong Youth Scholar Chair Professor of the Ministry of Education.

## Author contributions

J.Y. and M.Z. designed the research. Z.L. and G.Q. analyzed glucosinolate components. J.L. and J.Y. performed comparative genomic and evolutionary analyses. P.S., J.H., and Z.H. collected some data from the experiments. J.Y. wrote the paper. M.Z. revised the paper.

## Competing interests

The authors declare no competing interests.
