## [Peer Review File · Communications Biology]

Reviewers' Comments:

Reviewer #1:

Remarks to the Author:

Yang et al., report on sinigrin compounds analysis in Brassicaceae crops using RNA-Seq and, LC-MS HPLC analysis. The authors suggested convergent evolution and new candidate transporters of GSLs. The author used de novo transcriptome assembly to characterize genes in *W. japonica*, *A. rusticana* and *B. juncea*. The authors used two-electrode voltage-clamp measurements in *Xenopus laevis* oocyte, which preferentially uses for a transporter activities measurement. Undoubtedly, the RNA-seq results described for GSLs and sinigrin compounds are novel and have to potential to be of broad interest to the Brassica metabolism research community.

However, I have several major concerns about the current manuscript. The introduction could be improved with a clearly stated the rationale for conducting RNA-seq transcriptome profiling leaf and root for providing background information about the specific role that GTRs play in transporting sinigrin and glucobrassicin in Brassica. I also have several reservations about the quality-control validation of the RNA-Seq and transcriptome assembly. Furthermore, I also have issues with the K_a and K_s calculation with transcriptome assembly reported in the manuscript. More detailed comments follow below.

The introduction should be revised to set the stage about GSL transport, the role of GTRs behavior as indole-specific GSLs transporter, and to correct some errors. Jørgensen et al., mentioned all three (GTR1-3) knock down shows increase in the rosette indole glucosinolate content, not only GTR3.

The statement that "AtNPF2.9 (AtGTR3) serves as the indole-specific GSLs transporter from leaves to roots" is incorrect as stated because of all three need to be knock down. Please revise the wording here.

For general citation regarding nitrate transport is good to add: Complex phylogeny and gene expression patterns of members of the NITRATE TRANSPORTER 1/PEPTIDE TRANSPORTER family (NPF) in wheat Buchner, P.; Hawkesford, M.J. (2014) *Journal of Experimental Botany*, Uptake, allocation and signaling of nitrate Wang, Y.Y.; Hsu, P.K.; Tsay, Y.F. (2012) *Trends in Plant Science*.

The author skipped to mention the relationship between sinigrin and the taste. "Class targeted metabolomics: ESI ion trap screening methods for glucosinolates based on MS n fragmentation Rochfort, S.J.; Trenerry, V.C.; Imsic, M.; Panozzo, J.; Jones, R. (2008) *Phytochemistry*" Also the statement line:81-83 should move to introduction.

Many previous studies have specifically examined the gene clusters in distinct evolutionary lineages. Perhaps the author could cite some of these earlier studies.

The authors used RNA-seq method and construct transcriptome assembly by de novo method. No quality control metrics or evidence of completeness were provided in the current study to validate the assembly results. I also have some major reservations about the transcriptome assembly methods used. For example, the authors used Trinity pipeline. In my expertise, Trinity tend to generate a lot of transcripts which includes alternative splicing in exon and UTR region. Authors did not specify any method to generate Unigene. In addition, the authors should provide some quality metrics for the *B. juncea* transcriptome assembly by comparing current genome data.

The authors used K_a and K_s calculation by PAML package. However, the completeness of transcriptome assembly will give a huge misleading in calculation (Fig. S4.)

I also have a major concerns about the convergent evolution of Brassica. Although Arabidopsis GTR4 and GTR5 orthologs can play essential roles in transporting sinigrin and glucobrassicin, this

peptide/nitrate transporter and glyoxal oxidase-related protein has been experience several duplications according to its evolutionary fate. Does PCOC handle this plant specific genome duplication scenario? This can be covered by additional Ka and Ks calculation. However I still have concern about incomplete transcripts assembly.

However, GTR4 and GTR5 functionalization are novel and have to potential to be of broad interest to the plant research community.

Reviewer #2:

Remarks to the Author:

The authors use wasabi, horseradish, and mustard to show convergent evolution in the well-studied specialized metabolism pathways of glucosinolate biosynthesis, in particular the biosynthesis of sinigrin. Also, this work identifies two previously unknown transporter genes, GTR4 and GTR5, and functionally characterizes them as transporters of the glucosinolates sinigrin and glucobrassicin. Finally, they show how it is possible to identify candidate genes for genetic engineering via convergent evolution. Overall this paper shows promise and would be of interest to those in both basic plant science and applied crop science.

Major concerns

1. Data availability is lacking in methods or supplement, making it difficult to evaluate the results. In the methods section the authors describe the major analysis of transcriptome assembly, Ka/Ks calculation, positive selection analysis and convergent evolution analysis, all of which are used in the results section to evaluate genes and claim positive selection or convergent evolution, however in the data tables I do not see any information on the actual values of each of these analyses. For example, in lines 125-125 the authors claim that some GSL related genes have higher expression in roots, however there are no actual values for log fold change or expression level in the supplemental tables, so there is no way for the reviewer to check if this is actually the case. Other examples include lines 153-154: "The results revealed 478 genes displaying to be under positive selection in *B. juncea*, *A. rusticana* and *W. japonica* (Supplementary Table 7)". When one looks at Supp. Table 7 however, we find the lists of genes that are under positive selection, but their ka/ks value is not included, so we cannot evaluate the claim that they are indeed under positive selection. The same can be said for the convergent evolution analysis- while this analysis is very intriguing, we are lacking the values of the analysis for the genes that are discussed in the paper.

This paper would benefit from including all of the data from the analyses, or at least the data for the genes that are discussed in the paper. I would recommend that the authors include a table with log fold change or expression values, a table with the Ka/Ks values, and a table with convergent evolution values for all the genes in each species.

2. Crucial details in methods seem to be missing, although perhaps authors omitted some detail for lack of space, but if this is the case, please include a supplemental methods section. Details regarding the knockout experiment seem to be vague or incomplete. For lines 335-337, the authors describe the screening of a knockout mutant, but more information on this mutant is needed. How did you get the mutant, i.e. did you make it or order it? If it was ordered then where is it from (reference)?

Also, the authors refer to doing "gene expression analysis" both here and other places in the paper, but gene expression analysis is never described (although it is referenced). It would be helpful to have at least a brief explanation of how the gene expression analysis was done, and likely computational tools should be referenced here.

Minor comments:

3. Certain places in the results are vague. Examples include lines 197 "expressions did not change so much". It would be better to have an actual value here, or to reference a value. We don't know what not so much means. Another example are lines 124-125 "GSLs side chain modifications were higher expressed in roots". First, higher implies a comparison, but we don't know what that comparison is (roots compared to leaves?). Second, again a value is lacking- how much higher? Is it significantly higher? If so then a p-value is needed.

4. There are a couple of areas that should be re-worded. In lines 86-89 the authors say "sinigrin accumulation in these three species is considered as convergent evolution, because these three species are belonged to distinct evolutionary lineages". This causes confusion because the three species are all part of the Brassicaceae family, thus some could argue that they are not distinct. The authors should clarify what makes these lineages distinct.

5. Lines 73-74, Brassicaceae is a family not a genus.

6. Throughout: authors go between specialized and secondary when referring to this type of metabolism. Stick with one (probably specialized since this is the most recent consensus wording).

7. The section "Functional analysis of new candidate transporters of GSLs" would benefit from a pathway figure so we can see where the new candidate genes would fit in.

8. Line 145 and throughout: Abbreviations should be defined before used. In this line Ka/Ks has not been defined yet.

Reviewer #3:

Remarks to the Author:

This manuscript has unique and valuable information on the role of GSLs transporter(GTR)for sinigrin accumulation in edible tissues of wasabi, mustard, and horse radish by addressing convergent evolution. The results and discussion in the present study bring a better overview regarding translocation and accumulation of sinigrin with new GTRs from leaf to sink organs in Brassica crops.

However, the following comments should be clarified to support the conclusions shown in the abstract.

1. It is strongly recommended to provide convincing data/information in order to substantiate the synthesis/accumulation of sinigrin in leaf tissues such as AOP2 gene(s)since this gene is responsible to convert glucoiberin to sinigrin in several Brassica crops.

2. It would be clearer to provide comparative explanation on GSLs metabolic pathway not just transcriptome, especially side chain modifications between three Brassica crops of the present experiment and the other brassica vegetables.

3. Since the GSLs are widely diverse depends on stage and tissue/organs, it should be provided developmental stage of each organs and plant in Materials and Method. More detailed information is necessary on how to obtain seeds whether it from the same plant with roots and leaves or separate, considering sink and source relationship of the organs.

4. Fig.1b of GSLs content should provide the relative amount(%) as well as content of individual GSL.

Reviewer #4:

Remarks to the Author:

The manuscript hypothesizes that phylogenetically remote species that have evolved the same profile of GLSs may also have evolved convergent evolution of GLS transporters. Based on the different organs for three species to accumulate sinigrin the authors comparatively analyze NPF

transporters and identify two NPF transporters presumably involved in the accumulation of GLSs and call them new GLS transporters (GTRs). If true it would be novel, but the biochemical and genetic evidence to support this are not convincing.

Specific comments:

As a non-geneticist I find that the text is not very reader-friendly for the general reader. Also, the language is not very descriptive and results not explained.

l. 21: Impression in language, e.g. 'specialized glucosinolates' is that a particular glucosinolates or does the author refer to that glucosinolates are specialized metabolites?

l. 27: NRT/PTR family has been renamed in 2014 to NPF family and this name should be used today.

l.120-onward, the authors comments on the heatmaps that apparently shows that some identical functional genes (IGTM1, FMOs, and CYP81F1) are expressed in leaves of all species. Particularly (line 132-134), for NPF transporters the authors observe some hitherto unknown transporters that are highly expressed and thus the authors suggest that they are potent candidate GLS transporters. It is not clear how these findings are correlated with the heat maps, so the reader cannot judge any of these statements. Please provide more explanation. Since the findings of GTR4 and GTR5 is key in this paper, more information is needed to allow the reader to follow by themselves exactly how these new potential GLS transporters were found.

l.132. 'Besides higher expression of GTR1,-2 and -3' compared to what? Also, are transporters necessarily identified based on guilt-by-association?

l.185 How can two genes both be called GTR4 and the same for GTR5 (also fig .5)?

l.187 Uptake for 2 days in oocytes with so little error is unusual. How many oocytes were used per experiment?

What is the actual uptake in the oocytes e.g. compared to medium level? The relative value makes it not possible to judge. Making the activity relative to one of the transporters, but what if that one has very low uptake? If the authors insist on doing relative values it should be to one of the known GTRs.

Based on the existing data there is no evidence that these are GLS transporters. The knockout phenotype in Fig. S8) could be an indirect effect.

Also, why do the authors write that they do two-electrode voltage-clamp measurements. Did I miss some TEVC data?

Figure 1b. Let the area of the pizza represent the amount of total GLS in the given organ.

Figure 2b. The authors do not describe in results (around line 117) what they conclude from figure 2b.

Figure 2c. What is the purpose of this figure? It mixes three different processes: de novo synthesis, transport and breakdown. Why is 4OHB highlighted?

Fig5. Legend insufficient. Nothing about glucobrassicin.

Native English speaker should go through the manuscript for language.

Point-to-point response to reviewer comments for COMMSBIO-20-0763

Reviewer #1 (Remarks to the Author):

Yang et al., report on sinigrin compounds analysis in Brassicaceae crops using RNA-Seq and, LC-MS HPLC analysis. The authors suggested convergent evolution and new candidate transporters of GSLs. The author used de novo transcriptome assembly to characterize genes in *W. japonica*, *A. rusticana* and *B. juncea*. The authors used two-electrode voltage-clamp measurements in *Xenopus laevis* oocyte, which preferentially uses for a transporter activities measurement. Undoubtedly, the RNA-seq results described for GSLs and sinigrin compounds are novel and have to potential to be of broad interest to the Brassica metabolism research community.

However, I have several major concerns about the current manuscript. The introduction could be improved with a clearly stated the rationale for conducting RNA-seq transcriptome profiling leaf and root for providing background information about the specific role that GTRs play in transporting sinigrin and glucobrassicin in Brassica. I also have several reservations about the quality-control validation of the RNA-Seq and transcriptome assembly. Furthermore, I also have issues with the Ka and Ks calculation with transcriptome assembly reported in the manuscript. More detailed comments follow below.

Authors' Response: Thanks for the comments. The glucosinolates are considered to be mainly synthesized in leaves, and then are transported to other organs by GTRs, like seeds, particularly for long-chain aliphatic GLSs (by Halkier group). In root, GTR-mediated import is essential for retention of GLSs, and in the *gtr1gtr2* mutant,

the GSLs are moved from root to shoot. In contrast, short-chain aliphatic GSLs were shown to be mainly produced in the leaves and transported to the roots. The extraordinary and interesting materials in the present studies, wasabi, horseradish and mustard, mainly accumulated aliphatic GSLs component, sinigrin, in seeds or roots. The diverse GSLs components are supposed to need different transporters in *Brassicaceae* crops and model plant *Arabidopsis*. So, we mainly investigated the transcriptional patterns in leaves, seeds and roots. We added such research background in the Introduction section.

We estimated transcriptome assembly using BUSCO in these three species and supplemented this data in this revised version, although it does not show high BUSCO with the genome assembly because of only a few tissues included (Supplementary Table 6). But, we believe that the data quality is adequate to be used to conduct comparative analyses including Ka/Ks calculation and the differences identified here are convincing together with functional analyses, although it might be not fully studied. For Ka/Ks calculation, we used all one-to-one orthologous genes to calculate based on the phylogenetic tree of seven plant species.

The introduction should be revised to set the stage about GSL transport, the role of GTRs behavior as indole-specific GSLs transporter, and to correct some errors. Jørgensen et al., mentioned all three (GTR1-3) knock down shows increase in the rosette indole glucosinolate content, not only GTR3.

Authors' Response: Thanks for the suggestions. We added some background on GTRs transport in the Introduction section as suggested.

The statement that “AtNPF2.9 (AtGTR3) serves as the indole-specific GSLs transporter from leaves to roots” is incorrect as stated because of all three need to be knock down. Please revise the wording here.

Authors' Response: Thanks for your correction. We revised this description to be ‘Furthermore, GTR1, GTR2 and GTR3 (AtNPF2.9) all contribute to the distributions of indole GSLs between root and shoot’.

For general citation regarding nitrate transport is good to add: Complex phylogeny and gene expression patterns of members of the NITRATE TRANSPORTER 1/PEPTIDE TRANSPORTER family (NPF) in wheat Buchner,P.; Hawkesford,M.J.(2014)Journal of Experimental Botany, Uptake, allocation and signaling of nitrate Wang,YY.;Hsu,P.K.;Tsay,Y.F.(2012)Trends in Plant Science.

Authors Response: Thanks for your advice. We cited the two related reference (35, 36) in this revised manuscript.

Buchner P, Hawkesford MJ. Complex phylogeny and gene expression patterns of members of the NITRATE TRANSPORTER 1/PEPTIDE TRANSPORTER family (NPF) in wheat. *J Exp Bot* **65**, 5697-5710 (2014).

Wang YY, Hsu PK, Tsay YF. Uptake, allocation and signaling of nitrate (vol 17, pg 458, 2012). *Trends Plant Sci* **17**, 624-624 (2012).

The author skipped to mention the relationship between sinigrin and the taste. “Class targeted metabolomics: ESI ion trap screening methods for glucosinolates based on MSⁿ fragmentation Rochfort,S.J.; Trenerry,V.C.; Imsic,M.; Panozzo,J.; Jones,R. (2008) *Phytochemistry*” Also the statement line:81-83 should move to introduction.

Authors’ Response: Thanks for your suggestion. We added relationship between sinigrin and taste in vegetable crops in the Introduction. And we translocate sinigrin description in the results (line:81-83) to the Introduction as suggested. Meanwhile, we added another related reference here including the mentioned one by reviewer (Ref. 12 and 13).

Rochfort SJ, Trenerry VC, Imsic M, Panozzo J, Jones R. Class targeted metabolomics: ESI ion trap screening methods for glucosinolates based on MSⁿ fragmentation. *Phytochemistry* 69, 1671-1679 (2008).

Cools K, Terry LA. The effect of processing on the glucosinolate profile in mustard seed. *Food Chem* 252, 343-348 (2018).

Many previous studies have specifically examined the gene clusters in distinct evolutionary lineages. Perhaps the author could cite some of these earlier studies.

Authors’ Response: We added description on gene clusters and glucosinolates metabolism in *Brassicaceae* supported by three references (Ref. 39, 40 and 41).

Nutzmann HW, Osbourn A. Gene clustering in plant specialized metabolism. *Curr Opin Biotech* 26,

91-99 (2014).

Kliebenstein DJ, Lambrix VM, Reichelt M, Gershenzon J, Mitchell-Olds T. Gene duplication in the diversification of secondary metabolism: Tandem 2-oxoglutarate-dependent dioxygenases control glucosinolate biosynthesis in arabidopsis. *Plant Cell* **13**, 681-693 (2001).

Essoh AP, Monteiro F, Pena AR, Pais MS, Moura M, Romeiras MM. Exploring glucosinolates diversity in Brassicaceae: a genomic and chemical assessment for deciphering abiotic stress tolerance. *Plant Physiol Bioch* **150**, 151-161 (2020).

The authors used RNA-seq method and construct transcriptome assembly by de novo method. No quality control metrics or evidence of completeness were provided in the current study to validate the assembly results. I also have some major reservations about the transcriptome assembly methods used. For example, the authors used Trinity pipeline. In my expertise, Trinity tend to generate a lot of transcripts which includes alternative splicing in exon and UTR region. Authors did not specify any method to generate Unigene. In addition, the authors should provide some quality metrics for the *B. juncea* transcriptome assembly by comparing current genome data.

Authors' Response: Thanks for your comments. We estimated transcriptome assembly using BUSCO in these three species and supplemented this data in this revised version. To date, *B. juncea* reference genome was only published by our group, and we can only compare transcriptome assembly and genome assembly in *B. juncea*. Because the *B. juncea* variety (for vegetable purpose) used for genome

assembly and transcriptome assembly (for condiment purpose) are different, we can not use genome assembly for comparative analysis in this study. Although it does not show high BUSCO with the genome assembly because of only a few tissues included (Supplementary Table 6), we think the genes we identified in this study are still convincing together with supporting data from functional analyses.

We added transcriptome assembly description in the Method section, which is belonged to a routine pipeline for transcriptome assembly and unigene identification. Transcriptome *de novo* assembly was carried out with the publicly available program Trinity. The following parameters were used in Trinity: min_glue=3, V=10, edge-thr=0.05, min_kmer_cov=3, path_reinforcement_distance=85, group_pairs_distance=250, and the other parameters were set as the default. Next, any redundant fragments were removed by TGICL (TGI Clustering tools) and Phrap assembler. The following parameters were used to ensure a high quality of assembly: a minimum of 95% identity, a minimum of 35 overlapping bases, a minimum of 35 scores and a maximum of 25 unmatched overhanging bases at sequence ends.

We only selected the one to one orthologous genes among seven Brassicaceae plants (*A. rusticana*, *A. thaliana*, *B. juncea*, *B. napus*, *B. oleracea*, *B. rapa*, *W. japonica*) for positive selection and convergent evolution analysis. The one-to-one orthologous genes were defined using BLASTP based on the Bidirectional Best Hit (BBH) method, followed by selection of the best match, a total of 4,496 one-to-one orthologous gene sets were found among the seven Brassicaceae plants.

The authors used Ka and Ks calculation by PAML package. However, the completeness of transcriptome assembly will give a huge misleading in calculation (Fig. S4.)

Authors' Response: Thanks for your comments. Although it does not show high BUSCO with the genome assembly because of only a few tissues included (Supplementary Table 6), we believe that the data can be used to comparative analyses including Ka/Ks calculation and the differences identified here are convincing combined with functional analyses, although it might be not fully studied. For Ka/Ks calculation, we used all one-to-one orthologous genes to calculate based on the phylogenetic tree of seven plant species.

I also have a major concerns about the convergent evolution of Brassica. Although Arabidopsis GTR4 and GTR5 orthologs can play essential roles in transporting sinigrin and glucobrassicin, this peptide/nitrate transporter and glyoxal oxidase-related protein has been experience several duplications according to its evolutionary fate. Does PCOC handle this plant specific genome duplication scenario? This can be covered by additional Ka and Ks calculation. However I still have concern about incomplete transcripts assembly.

Authors' Response: Thanks for your comments. In PCOC analysis, we used all one-to-one orthologous genes to identify candidate genes with convergent amino acid shiftings. The duplications genes in species could be identified to be one-to-one orthologs according to sequence similarity.

However, GTR4 and GTR5 functionalization are novel and have to potential to be of broad interest to the plant research community.

Authors' Response: Thanks for your comments. We added new data, from GTR4 and GTR5 structural perspectives, on sinigrin transporter substrate analysis using homology modeling method referred to Jørgensen et al., 2017 (eLife).

Reviewer #2 (Remarks to the Author):

The authors use wasabi, horseradish, and mustard to show convergent evolution in the well-studied specialized metabolism pathways of glucosinolate biosynthesis, in particular the biosynthesis of sinigrin. Also, this work identifies two previously unknown transporter genes, GTR4 and GTR5, and functionally characterizes them as transporters of the glucosinolates sinigrin and glucobrassicin. Finally, they show how it is possible to identify candidate genes for genetic engineering via convergent evolution. Overall this paper shows promise and would be of interest to those in both basic plant science and applied crop science.

Major concerns

1. Data availability is lacking in methods or supplement, making it difficult to evaluate the results. In the methods section the authors describe the major analysis of transcriptome assembly, Ka/Ks calculation, positive selection analysis and convergent evolution analysis, all of which are used in the results section to evaluate genes and claim positive selection or convergent evolution, however in the data tables I do not see any information on the actual values of each of these analyses.

For example, in lines 125-125 the authors claim that some GSL related genes have higher expression in roots, however there are no actual values for log fold change or expression level in the supplemental tables, so there is no way for the reviewer to check if this is actually the case. Other examples include lines 153-154: “The results revealed 478 genes displaying to be under positive selection in *B. juncea*, *A. rusticana* and *W. japonica* (Supplementary Table 7)”. When one looks at Supp. Table 7 however,

we find the lists of genes that are under positive selection, but their ka/ks value is not included, so we cannot evaluate the claim that they are indeed under positive selection. The same can be said for the convergent evolution analysis- while this analysis is very intriguing, we are lacking the values of the analysis for the genes that are discussed in the paper.

This paper would benefit from including all of the data from the analyses, or at least the data for the genes that are discussed in the paper. I would recommend that the authors include a table with log fold change or expression values, a table with the Ka/Ks values, and a table with convergent evolution values for all the genes in each species.

Authors' Response: Thanks for your comments. We deposited the transcriptome assembly data in CNSA (<https://db.cngb.org/cnsa/>) under accession number CNP0001119. Meanwhile, as suggested, we added a Supplementary Tables 8 for transcriptional expression levels of glucosinolates metabolism related genes in Figure 2d-f and Ka/Ks values of positive selected genes in Supplementary Table S9 in the revised MS. Furthermore, we added a Supplementary Table 10 for convergent shiftings of amino acid sites in convergent evolution analyses of genes involving in glucosinolates metabolism in Figure 4.

2. Crucial details in methods seem to be missing, although perhaps authors omitted some detail for lack of space, but if this is the case, please include a supplemental methods section. Details regarding the knockout experiment seem to be vague or

incomplete. For lines 335-337, the authors describe the screening of a knockout mutant, but more information on this mutant is needed. How did you get the mutant, i.e. did you make it or order it? If it was ordered then where is it from (reference)?

Also, the authors refer to doing “gene expression analysis” both here and other places in the paper, but gene expression analysis is never described (although it is referenced). It would be helpful to have at least a brief explanation of how the gene expression analysis was done, and likely computational tools should be referenced here.

Authors’ Response: Thanks for your valuable comments. We added detailed description on *Arabidopsis* knockout mutants in Methods section.

The SALK_080802C (NASC code, N679871) and SALK_090216C (NASC code, N657255) lines were ordered from AraShare (www.arashare.cn) and NASC (<http://arabidopsis.info/>), and were used for screening of homozygous *at5g13400* and *at3g53950* knockout-out mutants of *Arabidopsis* using the three primers sets designed from the online service (<http://signal.salk.edu/tdnaprimers.2.html>), respectively. Seeds and roots samples from *at5g13400* and *at3g53950* knockout mutants and wild type of *Arabidopsis* were collected for GSLs components⁴¹ and gene expressions analysis using quantitative Real-time PCR analysis. Total RNAs were extracted from investigated samples using RNAprep Pure Plant Kit (TIANGEN, Beijing, China). RNase-free DNase I (TIANGEN, Beijing, China) exhaustively digested the total DNAs during extraction process. 1 µg total RNAs were then reversely transcribed to cDNA using a ReverTra Ace qPCR RT Master Mix with gDNA Remover (TOYOBO,

Japan). The relative transcriptional expressions of interested genes were analyzed by quantitative real-time PCR on ABI Step One Plus (Applied Biosystems, USA). The relative transcriptional quantification was determined using $\Delta\Delta$ CT method. All analysis was run in three independent biological replicates. All primers used in this study were listed in Supplementary Table 11.

Minor comments:

3. Certain places in the results are vague. Examples include lines 197 “expressions did not change so much”. It would be better to have an actual value here, or to reference a value. We don’t know what not so much means. Another example are lines 124-125 “GSLs side chain modifications were higher expressed in roots”. First, higher implies a comparison, but we don’t know what that comparison is (roots compared to leaves?). Second, again a value is lacking- how much higher? Is it significantly higher? If so then a p-value is needed.

Authors’ Response: Sorry for this inaccurate description. We revised ‘expressions did not change so much’ to ‘obvious changes in GTR1 and GTR2 expressions were observed’. In addition, we changed ‘GSLs side chain modifications were higher expressed in roots’ to ‘GSLs side chain modifications were higher expressed in roots compared to those with similar expression patterns in leaves, which suggested that GSLs might also be synthesized in root’.

4. There are a couple of areas that should be re-worded. In lines 86-89 the authors

say “sinigrin accumulation in these three species is considered as convergent evolution, because these three species are belonged to distinct evolutionary lineages”. This causes confusion because the three species are all part of the Brassicaceae family, thus some could argue that they are not distinct. The authors should clarify what makes these lineages distinct.

Authors’ Response: Sorry for causing confusion. We rewrote this sentence to ‘We constructed a phylogenetic tree among wasabi, horseradish and mustard species with several other sequenced species using one-to-one orthologs in the *Brassicaceae* family. The wasabi, horseradish and mustard species are belonged to different lineages in the family (Fig. 1a). Consequently, they underwent convergent evolution in sinigrin accumulation with similar phenotypes in distinct evolutionary lineages of the same family’.

5. Lines 73-74, Brassicaceae is a family not a genus.

Authors’ Response: Sorry for this mistake. We revised it as a family.

6. Throughout: authors go between specialized and secondary when referring to this type of metabolism. Stick with one (probably specialized since this is the most recent consensus wording).

Authors’ Response: Thanks for the comments. We substitute all ‘secondary’ wording with ‘specialized’.

7. The section “Functional analysis of new candidate transporters of GSLs” would benefit from a pathway figure so we can see where the new candidate genes would fit in.

Authors’ Response: Thanks for the suggestions. We added a schematic figure in Figure 5f to show their potential roles in sinigrin transport. We also added GTR4 and GTR5 subcellular localization data to show their targeting to the plasma membrane in Figure 5d.

8. Line 145 and throughout: Abbreviations should be defined before used. In this line Ka/Ks has not been defined yet.

Authors’ Response: We added full names for Ka/Ks (non-synonymous/synonymous) when they were firstly used in the text.

Reviewer #3 (Remarks to the Author):

This manuscript has unique and valuable information on the role of GSLs transporter (GTR) for sinigrin accumulation in edible tissues of wasabi, mustard, and horse radish by addressing convergent evolution. The results and discussion in the present study bring a better overview regarding translocation and accumulation of sinigrin with new GTRs from leaf to sink organs in Brassica crops.

However, the following comments should be clarified to support the conclusions shown in the abstract.

1. It is strongly recommended to provide convincing data/information in order to substantiate the synthesis/accumulation of sinigrin in leaf tissues such as AOP2 gene(s) since this gene is responsible to convert glucoiberin to sinigrin in several *Brassica* crops.

Authors' Response: Thanks for the comments. We supplemented AOP2s expressions in leaves of these three species as Supplementary Figure 4, in which AOP2s expressions showed significantly higher in leaves than in seeds, and higher expression of AOP2 in roots in horseradish might indicate that sinigrin could be synthesized in roots.

2. It would be clearer to provide comparative explanation on GSLs metabolic pathway not just transcriptome, especially side chain modifications between three Brassica crops of the present experiment and the other brassica vegetables.

Authors' Response: Thanks for the comments. We added comparative explanations

in on side chain modifications between these three species with other *Brassica* vegetables in the *Brassicaceae* family in discussion section (Line 271-274).

3. Since the GSLs are widely diverse depends on stage and tissue/organs, it should be provided developmental stage of each organs and plant in Materials and Method. More detailed information is necessary on how to obtain seeds whether it from the same plant with roots and leaves or separate, considering sink and source relationship of the organs.

Authors' Response: Thanks for the suggestions. We added detailed descriptions on samples collections in developmental stages of each organ in Materials and Methods part.

We collected leaves and roots samples at roots organs formation and harvested seeds from different plants when the seeds were mature in the populations.

4. Fig.1b of GSLs content should provide the relative amount (%) as well as content of individual GSL.

Authors' Response: Thanks for the suggestions. We added the relative amount (%) and true value of sinigrin in these tissues and others relative proportions in Supplementary Table 2.

Reviewer #4 (Remarks to the Author):

The manuscript hypothesizes that phylogenetically remote species that have evolved

the same profile of GLSs may also have evolved convergent evolution of GLS transporters. Based on the different organs for three species to accumulate sinigrin the authors comparatively analyze NPF transporters and identify two NPF transporters presumably involved in the accumulation of GLSs and call them new GLS transporters (GTRs). If true it would be novel, but the biochemical and genetic evidence to support this are not convincing.

Authors' Response: Thanks for the comments. For the new candidate GTRs functional analysis, firstly, we employed *in vitro* functional analysis of new candidate GTRs using *X. laevis* oocytes. Next, we checked the functions of the orthologs of new candidate GTRs in *Arabidopsis*. These two complementary methods indicated that the new candidate GTRs have transport activities for sinigrin or other GSLs components.

Specific comments:

As a non-geneticist I find that the text is not very reader-friendly for the general reader. Also, the language is not very descriptive and results not explained.

Authors' Response: Thanks for the comments. We made English language editing from English native professional editing service in this revised version to make it more readable and easily understood.

l. 21: Impression in language, e.g. 'specialized glucosinolates' is that a particular glucosinolates or does the author refer to that glucosinolates are specialized

metabolites?

Authors' Response: Glucosinolates are specifically synthesized in *Brassicaceae* and other few families in plants. In some review papers, they named such kind of metabolites as specialized metabolites, if the metabolites are particularly enriched in this species. Here, specialized glucosinolates means that glucosinolates are specialized metabolites in *Brassicaceae* species.

I. 27: NRT/PTR family has been renamed in 2014 to NPF family and this name should be used today.

Authors' Response: Thanks for your suggestions. We used 'NRT/PTR family' instead of 'NPF family' in the text.

I.120-onward, the authors comments on the heatmaps that apparently shows that some identical functional genes (IGTM1, FMOs, and CYP81F1) are expressed in leaves of all species. Particularly (line 132-134), for NPF transporters the authors observe some hitherto unknown transporters that are highly expressed and thus the authors suggest that they are potent candidate GLS transporters. It is not clear how these findings are correlated with the heat maps, so the reader cannot judge any of these statements. Please provide more explanation. Since the findings of GTR4 and GTR5 is key in this paper, more information is needed to allow the reader to follow by themselves exactly how these new potential GLS transporters were found.

Authors' Response: Thanks for your comments. We firstly used orthologs of the

three species from *Arabidopsis* to check their expression patterns and we included a new Supplementary Table S8 with all these genes expressions levels. We can find differential expressed genes involved in GSLs metabolism with gene model annotation in *Arabidopsis* combined with Supplementary Table S7. From the expression patterns, we observed some new orthologs displaying high expression levels besides previously known GTR1, GTR2 and GTR3. So, we hypothesized that new glucosinolates transporters probably existing in these three species can specifically transport sinigrin. Next, we employed positive selection genes and convergent evolution analyses to identify candidate genes involving in glucosinolates metabolism under selection or convergent evolution in *Brassicaceae*. Besides new transporters, we are particularly interested in the glucosinolates transport process, because glucosinolates transport shapes flavor quality in seeds and at least partially in roots. We then checked the new candidate GTRs functions.

I.132. 'Besides higher expression of GTR1,-2 and -3' compared to what? Also, are transporters necessarily identified based on guilt-by-association?

Authors' Response: Thanks for the comments, we revised it as 'Besides higher expressions of *GTR1*, *GTR2* and *GTR3* in leaves, roots and seeds of the three species, we meanwhile observed some genes, in the NRT/PTR family displaying higher expression patterns in these tissues.

I.185 How can two genes both be called GTR4 and the same for GTR5 (also fig .5)?

Authors' Response: We are sorry for this mistake. We corrected them as GTR4 and GTR5.

I.187 Uptake for 2 days in oocytes with so little error is unusual. How many oocytes were used per experiment? What is the actual uptake in the oocytes e.g. compared to medium level? The relative value makes it not possible to judge. Making the activity relative to one of the transporters, but what if that one has very low uptake? If the authors insist on doing relative values it should be to one of the known GTRs.

Authors' Response: We are sorry for this mistake in Method section. It should be uptake for 60 min in oocyte. We corrected the method description as following and showed the actual sinigrin uptake value in Fig. 5b in this MS.

The injected oocytes (15 oocytes) were incubated at 16°C in Kulori solution (90 mM NaCl, 1 mM KCl, 1 mM MgCl₂, 10 mM MES) at pH 7.4 for two days. Then oocytes were pre-incubated in Kulori pH 5 for 5 min, then transferred to Kulori pH 5 with sinigrin (500 μM) and glucobrassicin (400 μM) for 60 min incubation respectively, followed by four washes and transferred to GEB centrifuge tubes (5 oocytes per tube). Desulfo glucosinolate was analyzed by HPLC, and standard curve was made by sinigrin (Solarbio, Beijing, China) to quantify the glucosinolates.

Based on the existing data there is no evidence that these are GLS transporters. The knockout phenotype in Fig. S8) could be an indirect effect.

Authors' Response: Actually, from Fig. S8 (Fig. S10 in this new version), we can see

that GTR3 is induced in *gtr4* and *gtr5* ortholog (*at5g13400* and *at3g53950*) mutants, of which GTR3 can transport indolic GSLs to roots. Though GTR3 expressions are induced (known function), the glucosinolates contents are decreased in *gtr4* and *gtr5* ortholog (*at5g13400* and *at3g53950*) mutants (Figure 5d). This result supports that GTR4 and GTR5 play roles of transporting GSLs to roots.

Also, why do the authors write that they do two-electrode voltage-clamp measurements. Did I miss some TEVC data?

Authors' Response: We are sorry for this mistake. We don't have TEVC data in this MS and only have uptake data. We deleted the 'two-electrode voltage-clamp measurements' and rewrote as 'We subsequently checked the *in vitro* transport activity of the two candidate GSLs transporter genes from *B. juncea* using heterogenous expressions in *X. laevis* oocytes.'

Figure 1b. Let the area of the pizza represent the amount of total GLS in the given organ.

Authors' Response: Thanks for the suggestions. We added the relative amount (%) and true value of sinigrin in the tissues and others relative proportions in Supplementary Table 2.

Figure 2b. The authors do not describe in results (around line 117) what they conclude from figure 2b.

Authors' Response: Thanks for your comments. We added description on Figure 2b in this revised version.

The biological replications of transcriptome sequencing were validated by the principal component analysis in leaves, roots and seeds of three species indicating that expression patterns have high similarities between biological replications (Fig. 2b).

Figure 2c. What is the purpose of this figure? It mixes three different processes: de novo synthesis, transport and breakdown. Why is 4OHB highlighted?

Authors' Response: In Fig. 2c, we displayed the three main processes of *de novo* glucosinolates synthesis, transport and breakdown, and sinigrin is predominant component. In this study, we mainly focused on sinigrin accumulation in these three species. So, we virtualized the other GSLs components pathway in Fig. 2c, in which 4OHB is belonged to indolic GSLs.

Fig5. Legend insufficient. Nothing about glucobrassicin.

Authors' Response: Sorry for this insufficient description in Figure 5. We added Figure legend for glucobrassicin in Fig. 5b.

Native English speaker should go through the manuscript for language.

Authors' Response: As suggested, we have asked professional agency to make language editing throughout the manuscript.

Reviewers' Comments:

Reviewer #1:

Remarks to the Author:

Author declared all the points in rebuttal letter.

Reviewer #2:

Remarks to the Author:

The authors have done a great job in addressing my major and minor concerns with a good attention to detail. I think adding in tables that show calculated expression levels and k_a/k_s values has greatly increased the transparency of the study. With no further concerns, I recommend this study for publication and I look forward to seeing the study in print.

Reviewer #3:

Remarks to the Author:

In Materials and Methods, developmental stage of each plant should be described such as days after germination and growth condition of those plant should also be depicted.

In Fig 1 and Table S2, unit of glucosinolates should be included in the pie graph. In addition, the size of each pie can be arranged according to the total glucosinolates content of each organ.

Authors can refer Fig. 1 and Table 2 from the following report:

Brown PD, Tokuhsa JG, Reichelt M, Gershenzon J (2003) Variation of glucosinolate accumulation among different organs and developmental stages of *Arabidopsis thaliana*. *Phytochemistry* 62:471–481. [https://doi.org/10.1016/S0031-9422\(02\)00549-6](https://doi.org/10.1016/S0031-9422(02)00549-6)

Reviewer #4:

Remarks to the Author:

see attached

In my first review, I argued for that this paper did not provide evidence that the GTR4 and GTR5 are glucosinolate transporters. The authors argue that their oocyte data knockout data support the claim.

I will in the following argue for that this manuscript should not be published as a story about two genes named GTR4 and GTR5 as I do not think there is evidence to support this and it will seriously confuse the community.

LONG-DISTANCE TRANSPORT PREMISE NOT CORRECT

There is premise made on page 6 line 111 that is not correct, and that is oversimplifying conclusions from the quoted refs (all published from my group).

GSLs, particularly aliphatic GSLs, have been shown to be synthesized in leaves, and then translocated from source to sink by transporters^{20, 21, 25}.

GLS are synthesized in many different tissues (root, leaf, stem, silique wall – everywhere but seeds), also in the three crops. Thus, the premise, that the leaves are the GLS factory of brassicaceous plants from which the GLS are moved to the final sink destination is incorrect.

These results suggested that sinigrin shapes the quality of wasabi in these three species. We believe that this occurs via a diverse range of transporters, in which mustard utilizes seed-specific transporters, horseradish uses a root-specific transporter, and wasabi utilizes both seed- and/or root-specific transporters. In addition, the super-accumulation of sinigrin in the seeds of mustard and wasabi, and in the roots of wasabi and horseradish, provide a good paradigm to comparatively investigate the diverse array of mechanisms used to transport GSLs.

The premise that e.g. mustard use seed-specific transporter to accumulate high levels in the seeds is oversimplifying. The route from source to sink requires transporters at every barrier along the way – not only transporters at the final destination.

Moreover, the gene discovery done in Arabidopsis with only two genes cannot easily be transferred to *B. juncea*. For each Arabidopsis GTR1 and GTR2 gene there are 6 homologs in *B. juncea* due to the triplication and hybridization occurring upon divergence of Arabidopsis and Brassica. For GTR1 and GTR2 - as seen in Nour-Eldin et al, Nat Biotech 2017 - these genes have very different expression patterns, are all functional alleles and many of the florally-expressed ones contribute to seed levels. I don't know what it is like in wasabi/horseradish.

GENE NAMING

I do not understand how the authors can arrive at the conclusion the GTR5 is a NPF transporter. According to Aramemnon database (aramemnon.botanik.uni-koeln.de), it has two membrane spanning domains (and not 12 as the NPFs). How a 2TMD glyoxal oxidase shows up in the phylogenetic tree stated to be a tree of the NPF family (fig S7) is a puzzle. Clearly, this cannot be a phylogenetic tree of the NPF family.

The At5g13400 is NPF6.1 and should be called this. Calling it GTR4 is confusing, particularly as there is no evidence to support this (see below).

LACK OF EVIDENCE FOR At GENES BEING GLS TRANSPORTERS

There is no biochemical evidence to support that Arabidopsis NPF6.1 or the 2TMD glyoxal oxidase are GLS transporters. The uptake assays are done on *B. juncea* genes, no data on Arabidopsis genes. It would be key to characterize the Arabidopsis genes before going to orthologs in crops. In this manuscript, Arabidopsis expression data and mutant data are combined with biochemistry on crop orthologs.

Measurement of (only) seed and root GLS levels of the knockouts of NPF6.1 and the glyoxal oxidase (Figure 5c,d) does not substantiate a conclusion that these genes are GLS transporters. There can be SO many reasons for altered GLS levels in these mutants.

This expression data of GTR1, GTR2, and GTR3 in root and seeds of the knockouts of NPF6.1 and the glyoxal oxidase (fig.S10) do not substantiate such a conclusion about the function of NPF6.1 and the glyoxal oxidase. The fact that GTR3 is highly upregulated in the both mutants can have many explanations. By calling these genes belonging to different gene families for GTR4 and GTR5, the reader may think that they are related somehow, which there is no evidence for.

BTW, it is incorrect to state that GTR3 is an indole transporter. In the paper characterizing GTR3 (Jørgensen ME et al. elife 2017), it was shown that GTR3 prefers indole over aliphatic GLS. GTR3 can also take aliphatic glucosinolates.

In summary, the authors use gene expression data (Fig. S10) together with the GLS data from knockout of NPF6.1 and the glyoxal oxidase to argue that these genes are GLS transporters (see reply to reviewer#4). However, there is no evidence to support that these Arabidopsis genes transport GLS. So, we should not confuse the community by calling these genes GTR4 and GTR5.

BRASSICA GENES EXPRESSION

The authors identify two 'GTR4' and two 'GTR5' homologs in mustard (in fig. 5b it now looks like there is 4 GTR4s, I guess the bottom two genes are the *Bj* glyoxal oxidases). Page 8 line 156 onwards it says:

***Previous study has shown that GTR1 and GTR2 are essential for the transport of both aliphatic and indole GSLs from leaves to seeds*^{21, 22},**

Ref 22 was published (Nour-Eldin et al, 2012). Since then, we have shown that the reason for the high levels of GLS in leaves of *gtr1 gtr2* mutant in Arabidopsis is not because the GLS did not get translocated to the seeds, but because (primarily long-chained aliphatic) GLS were not retained by the GTR1 and GTR2 in the root (ref 25, Madsen et al., Plant Phys. 2014). Thus, the root-derived GLS went via the xylem to accumulate in the leaves. Accordingly, this statement from Arabidopsis is not correct, and also it cannot be translated to the *B. juncea* that has a total of 12 genes, not two like Arabidopsis.

and that GTR1, GTR2 and GTR3 transport indole GSLs²⁵. Apart from the higher expression levels of GTR1, GTR2 and GTR3 in some leaves, roots and seeds of the three species, we observed some genes in the NPF that exhibited high expression levels in these tissues, thus indicating that these were potential transporters of GSLs (Fig. 2f).

There are 6 GTR1 and 6 GTR2 homologs in *B. juncea*. Is the sentence about expression levels of GTR1, GTR2 and GTR3 taking into consideration that we are talking about multiple genes? Moreover, I do not understand how one can pick a high-expressing NPF or glyoxal oxidase and use that as criterium for suggesting that these are GLS transporters (fig.2f).

When my group looked at the 6 orthologs of GTR1 and GTR2 in *B. juncea*, we found that these genes have very different expression patterns (Nour-Eldin H et al, Nat Biotech, 2017). In fig. S8, expression analysis is given for GTR1, GTR2, GTR3, GTR4, GTR5 in *B. juncea* (and also for wasabi and horseradish). Which GTR1 and which GTR2 orthologs from *B. juncea*? Maybe there is also 6 alleles of NPF6.1 and the glyoxal oxidase orthologs? So, what are we looking at? Is it an average of the expression of all the genes?

BIOCHEMISTRY

The authors have not included GTR1 or GTR2 from *Arabidopsis* to compare to the activity of NPF6.1, but have in the revised manuscript provided absolute values (it was relative values in the original manuscript). With 500 μ M sinigrin (quite a high concentration) in medium, NPF6.1/glyoxal oxidase accumulates max ~400 pmol pr oocyte pr h, which with an approx. volume of 1 μ l pr oocyte is 400 μ M, i.e. below medium level. So, the two putative GLS transporters do not accumulate above medium level when incubated in high substrate concentration. In our experience, the latter can somehow influence the result.

Typically, GTR1, GTR2 uptake approx. 2000-2500 pmol pr h using a medium concentration of 200 μ M (4msb or glucobrassicin), i.e. the transporter accumulates high above medium level. As suggested in my first review, it would be critical to include either AtGTR1 or AtGTR2 uptake assays measured under the same conditions in the same lab, as a positive control and for comparison. We are happy to provide these genes.

CONCLUSION

In conclusion, this manuscript does not provide evidence that NPF6.1 and glyoxal oxidase are a GLS transporters, and should not be published as such. I do not exclude that NPF6.1 transporters can have low GLS transport activity, but it remains to be shown convincingly, particularly considering the apparent promiscuity of transporters.

Barbara Ann Halkier (assisted by Hussam Nour-Eldin)

Reviewers' comments:

Reviewer #1 (Remarks to the Author):

Author declared all the points in rebuttal letter.

Authors' Response: Thanks for reviewer's comments.

Reviewer #2 (Remarks to the Author):

The authors have done a great job in addressing my major and minor concerns with a good attention to detail. I think adding in tables that show calculated expression levels and k_a/k_s values has greatly increased the transparency of the study. With no further concerns, I recommend this study for publication and I look forward to seeing the study in print.

Authors' Response: Thanks for reviewer's kind considerations.

Reviewer #3 (Remarks to the Author):

In Materials and Methods, developmental stage of each plant should be described such as days after germination and growth condition of those plant should also be depicted.

In Fig 1 and Table S2, unit of glucosinolates should be included in the pie graph. In addition, the size of each pie can be arranged according to the total glucosinolates content of each organ. Authors can refer Fig. 1 and Table 2 from the following report:

Brown PD, Tokuhsa JG, Reichelt M, Gershenzon J (2003) Variation of glucosinolate

accumulation among different organs and developmental stages of *Arabidopsis thaliana*. Phytochemistry 62:471–

481. [https://doi.org/10.1016/S0031-9422\(02\)00549-6](https://doi.org/10.1016/S0031-9422(02)00549-6)

Authors' Response: We added some description on plant growth stage and conditions in Materials and Methods in this revised version.

In addition, we revised the pie graph in Fig. 1b and Table S1 as suggested in this revised version.

Reviewer #4 (Remarks to the Author):

In my first review, I argued for that this paper did not provide evidence that the GTR4 and GTR5 are glucosinolate transporters. The authors argue that their oocyte data knockout data support the claim.

I will in the following argue for that this manuscript should not be published as a story about two genes named GTR4 and GTR5 as I do not think there is evidence to support this and it will seriously confuse the community.

1. LONG-DISTANCE TRANSPORT PREMISE NOT CORRECT

There is premise made on page 6 line 111 that is not correct, and that is oversimplifying conclusions from the quoted refs (all published from my group).

GSLs, particularly aliphatic GSLs, have been shown to be synthesized in leaves, and then translocated from source to sink by transporters^{20, 21, 25}.

GLS are synthesized in many different tissues (root, leaf, stem, silique wall – everywhere but seeds), also in the three crops. Thus, the premise, that the leaves are the GLS factory of brassicaceous plants from which the GLS are moved to the final sink destination is incorrect.

*These results suggested that sinigrin shapes **the quality of wasabi** in these three species. **We believe that this occurs via a diverse range of transporters**, in which mustard **utilizes seed-specific transporters**, horseradish **uses a root-specific transporter**, and wasabi **utilizes both seed- and/or root-specific transporters**. **In addition, the super-accumulation of sinigrin in the seeds of mustard and wasabi, and in the roots of wasabi and horseradish, provide a good paradigm to comparatively investigate the diverse array of mechanisms used to transport GSLs.***

The premise that e.g. mustard use seed-specific transporter to accumulate high levels in the seeds is oversimplifying. The route from source to sink requires transporters at every barrier along the way – not only transporters at the final destination.

Moreover, the gene discovery done in Arabidopsis with only two genes cannot easily be transferred to B. juncea. For each Arabidopsis GTR1 and GTR2 gene there are 6 homologs in B. juncea due to the triplication and hybridization occurring upon divergence of Arabidopsis and Brassica. For GTR1 and GTR2 - as seen in Nour-Eldin et al, Nat Biotech 2017 - these genes have very different expression patterns, are all functional alleles and many of the florally-expressed ones contribute to seed levels. I don't know what it is like in wasabi/horseradish.

2. GENE NAMING

I do not understand how the authors can arrive at the conclusion the GTR5 is a NPF transporter. According to Aramemnon database (aramemnon.botanik.uni-koeln.de), it has two membrane spanning domains (and not 12 as the NPFs). How a 2TMD glyoxal oxidase shows up in the phylogenetic tree stated to be a tree of the NPF family (fig S7) is a puzzle. Clearly, this cannot be a phylogenetic tree of the NPF family.

The At5g13400 is **NPF6.1** and should be called this. Calling it GTR4 is confusing,

particularly as there is no evidence to support this (see below).

3. LACK OF EVIDENCE FOR At GENES BEING GLS TRANSPORTERS

There is no biochemical evidence to support that Arabidopsis NPF6.1 or the 2TMD glyoxal oxidase are GLS transporters. The uptake assays are done on B. juncea genes, no data on Arabidopsis genes. It would be key to characterize the Arabidopsis genes before going to orthologs in crops. In this manuscript, Arabidopsis expression data and mutant data are combined with biochemistry on crop orthologs.

Measurement of (only) seed and root GLS levels of the knockouts of NPF6.1 and the glyoxal oxidase (Figure 5c,d) does not substantiate a conclusion that these genes are GLS transporters. There can be SO many reasons for altered GLS levels in these mutants.

This expression data of GTR1, GTR2, and GTR3 in root and seeds of the knockouts of NPF6.1 and the glyoxal oxidase (fig.S10) do not substantiate such a conclusion about the function of NPF6.1 and the glyoxal oxidase. The fact that **GTR3 is highly upregulated** in the both mutants **can have many explanations.** By calling these genes belonging to different gene families for GTR4 and GTR5, the reader may think that they are related somehow, which there is no evidence for.

BTW, it is incorrect to state that GTR3 is an indole transporter. In the paper characterizing GTR3 (Jørgensen ME et al. elife 2017), it was shown that **GTR3 prefers indole over aliphatic GLS.** GTR3 can also take aliphatic glucosinolates.

In summary, the authors use gene expression data (Fig. S10) together with the GLS data from knockout of NPF6.1 and the glyoxal oxidase to argue that these genes are GLS transporters (see reply to reviewer#4). However, there is no evidence to support that these **Arabidopsis genes** transport GLS. So, we should not confuse the community by calling these genes GTR4 and GTR5.

4. BRASSICA GENES EXPRESSION

The authors identify two 'GTR4' and two 'GTR5' homologs in mustard (in fig. 5b

it now looks like there is 4 GTR4s, I guess the bottom two genes are the Bj glyoxal oxidases). Page 8 line 156 onwards it says:

Previous study has shown that GTR1 and GTR2 are essential for the transport of both aliphatic and indole GSLs from leaves to seeds^{21, 22},

Ref 22 was published (Nour-Eldin et al, 2012). Since then, we have shown that the reason for the high levels of GLS in leaves of *gtr1 gtr2* mutant in Arabidopsis is not because the GLS did not get translocated to the seeds, but because (primarily long-chained aliphatic) GLS were not retained by the GTR1 and GTR2 in the root (ref 25, Madsen et al., Plant Phys. 2014). Thus, the root-derived GLS went via the xylem to accumulate in the leaves. Accordingly, this statement from Arabidopsis is not correct, and also it cannot be translated to the *B. juncea* that has a total of 12 genes, not two like Arabidopsis.

and that GTR1, GTR2 and GTR3 transport indole GSLs²⁵. Apart from the higher expression levels of GTR1, GTR2 and GTR3 in some leaves, roots and seeds of the three species, we observed some genes in the NPF that exhibited high expression levels in these tissues, thus indicating that these were potential transporters of GSLs (Fig. 2f).

There are 6 GTR1 and 6 GTR2 homologs in *B. juncea*. Is the sentence about expression levels of GTR1, GTR2 and GTR3 taking into consideration that we are talking about **multiple genes**? Moreover, I do not understand how one can pick a high-expressing NPF or glyoxal oxidase and use that as criterium for suggesting that these are GLS transporters (fig.2f).

When my group looked at the 6 orthologs of GTR1 and GTR2 in *B. juncea*, we found that these genes have very different expression patterns (Nour-Eldin H et al, Nat Biotech, 2017). In fig. S8, expression analysis is given for GTR1, GTR2, GTR3, GTR4, GTR5 in *B. juncea* (and also for wasabi and horseradish). Which GTR1 and which GTR2 orthologs from *B. juncea*? Maybe there is also 6 alleles of NPF6.1 and the glyoxal oxidase orthologs? So, what are we looking at? Is it an

average of the expression of all the genes?

5. BIOCHEMISTRY

The authors have not included GTR1 or GTR2 from Arabidopsis to compare to the activity of NPF6.1, but have in the revised manuscript provided absolute values (it was relative values in the original manuscript). With 500 uM sinigrin (quite a high concentration) in medium, NPF6.1/glyoxal oxidase accumulates max ~400 pmol pr oocyte pr h, which with an approx. volume of 1 ul pr oocyte is 400 uM, i.e. below medium level. So, the two putative GLS transporters do not accumulate above medium level when incubated in high substrate concentration. In our experience, the latter can somehow influence the result.

Typically, GTR1, GTR2 uptake approx. 2000-2500 pmol pr h using a medium concentration of 200uM (4msb or glucobrassicin), i.e. the transporter accumulates high above medium level. As suggested in my first review, it would be critical to include either AtGTR1 or AtGTR2 uptake assays measured under the same conditions in the same lab, as a positive control and for comparison. We are happy to provide these genes.

CONCLUSION

In conclusion, this manuscript does not provide evidence that NPF6.1 and glyoxal oxidase are a GLS transporters, and should not be published as such. I do not exclude that NPF6.1 transporters can have low GLS transport activity, but it remains to be shown convincingly, particularly considering the apparent promiscuity of transporters.

Authors' Response: Thanks for the comments. We rewrote the ambiguous description in this revised version.

As for the candidate transporters, we agree with the comments that we need to provide more sound evidence elaborately supported by further genetic complementation experiments. Accordingly, we are planning to take away the GTR4

and GTR5 portions from the entire article and resubmit it to CB after correcting some false statements according to the comments. In the past years, we had been fascinated about the hyper-accumulation of aliphatic GSLs in traditional Chinese seed mustard and Wasabi in Asian countries and horseradish in European countries widely used as pungent condiments. We believe that the convergent evolution of GSLs accumulation in these three crops are of great interest to the community and appealing to more focused research on these three unique crops that are super-rich in GSLs.